# The Limiting of the Impact of Proxy Culture Wars by Religious Sensitivity: The Fight of Neo-Pentecostal Churches against LGBTQ Rights Organizations over Uganda's Future

**Martin Palecek [1,\*] and Tomas Tazlar [2]**

1    Department of Philosophy and Social Sciences, University of Hradec Kralove (UHK), 500 03 Hradec Kralove, Czech Republic

2    Department of Political Science, University of Hradec Kralove (UHK), 500 03 Hradec Kralove, Czech Republic; tomas.tazlar@uhk.cz

\*    Correspondence: martin.palecek@uhk.cz

**Abstract:** It has been argued recently that Uganda's sexual law should be interpreted as a part of gender power struggles, rather than in the original neo-colonial interpretation or as a result of structural changes and President Museveni's pragmatic policy. Based on our intensive fieldwork during the dry season in 2017, we argue that an understanding of this development as a combination of the US proxy culture wars–US cultural wars being fought worldwide-interacting with local religious sensitivity is more plausible. The "sexual law" is a product of the clash between conservatives and progressivists over Uganda's future. The Neo-Pentecostals—typically supported by conservative circles from the USA and Canada—stand against the influence of secular NGOs—mostly connected with the LGBTQ and progressivist circles from the USA and EU. However, the effect of international influence is limited due to religious sensitivity, shaped by local tradition. Uganda's people are not passive victims of any kind. They take an active part in the global contest between cultural progressivists and conservatives.

**Keywords:** Uganda; cultural wars; Anti-Homosexuality Bill; Neo-Pentecostals; NGO; LGBTQ; wild religious traditions

## 1. Introduction

As summarized by Vorhölter (Vorhölter 2017, p. 94), there is a fully shared agreement among scholars that the entire anti-LGBTQ attitude in Uganda has its roots in colonial times (Aldrich 2003; Epperecht 2008; Han and O'Mahoney 2014; Nyanzi and Karamagi 2015). According to these scholars, the last outburst of anti-LGBTQ violence (Watt 2009) was a product of a neo-colonial attempt by conservative Neo-Pentecostal representatives to increase their influence in Uganda (Kaoma 2009; Cheney 2012; Bompani and Brown 2015; Bompani 2017; Bompani and Valois 2016)[1]. Both Kaoma (2009) and Cheney (2012) claimed that Neo-Pentecostal activists used a neo-colonial vocabulary for their purpose. They blamed American progressivists (linked to sexual rights NGOs) for a homo-colonial effort to bring LGBTQ issues into Africa, in order to erode traditional African family values.

However, Vorhölter (2017) noted that the Anti-Homosexuality Bill passed by the Ugandan Parliament in 2009 (revoked in 2014 after significant international pressure) was only one part of a sexual law that has been dramatically changing in Uganda. The Ugandan Parliament also passed the HIV Prevention and Control Act (2014), the Anti-Pornography Act (2014), Defilement, and Prostitution law[2]. Most recently, there has been an ongoing struggle concerning the Marriage (2017) (Marriage 2017; Activists Renew Demand for Marriage and Divorce Bill 2016). Vorhölter (2017) claims that the entire issue of passing the sexual law is, in fact, a patrimonial reaction to the changing gender roles in present day Uganda.

Based on our findings, we argue that, despite Vorhölter's insightful analysis, the original idea to interpret the sexual law as a part of Neo-Pentecostal influence and the struggle between conservative and progressive values is still more plausible. However, Vorhölter has a point in arguing that to interpret the present development in Uganda with an accent on the conservative Neo-Pentecostal lobby only (Kaoma 2009; Cheney 2012) is lacking in complexity, since it blocks an understanding of the Anti-Homosexuality Bill as a part of the whole cluster of sexual laws[3].

Moreover, explaining the development of the sexual laws as a mere result of structural changes and ideology (Bompani 2018; Bompani and Valois 2016) is not persuasive enough. It presupposes that ideology, in combination with the country's economic transition, is sufficient in order to manipulate public opinion. However, studies show that there are significant limits to people's gullibility (Mercier 2017, 2020). Even the most advanced ideology fails if human cognition is not sufficiently attracted (Boyer 2020a; Mercier 2020; Mercier and Sperber 2017)[4].

We argue that despite the insightfulness of the analyses mentioned above, we need to include the study of religious sensitivity and the role of NGOs in the conflict in order to gain a better understanding of present-day Uganda.

We propose the neologism proxy culture wars for the cultural development in Uganda. We have borrowed the term "proxy wars" from political science. This concept was initially assigned to several locally isolated international conflicts fought during the Cold War. We argue for the application of the concept of proxy wars together with culture wars because—as we discuss—US culture wars are being fought internationally.

Alongside proxy culture wars, we propose the concept of religious sensitivity to enhance the theory of religious attraction (Shariff et al. 2016; Peedu 2016; Parren 2017; Lawson and McCauley 1993; Connor Wood and Shaver 2018). We use the concept of religious sensitivity to emphasize those evolved abilities of the human mind that are intuitively triggered by religious marketing and are currently being exacerbated in combination with the local environment in present-day Uganda.

In other words, religious sensitivity is an evolved ability of the human mind to intuitively follow allegedly supernatural visions (in a broad sense), shaped according to local tradition and cultural history. We claim that it is this attractiveness that is responsible for the rapid success of Neo-Pentecostal churches and their agenda in present day Uganda (and not only there).

Our preliminary hypothesis was that the Ugandan sexual laws were a product of proxy culture wars, combined with local religious sensitivity: US cultural wars are being fought worldwide and are attractive enough to enable local Neo-Pentecostal churches to formulate an independent normative praxis that fits into local religious receptivity. Our findings fully supported this original hypothesis.

Our analysis brings us evidence that both wings of US culture wars were backing different sides of the conflict. In many details, members of religious and secular NGOs are under the influence of conservative or progressivist US circles, and they are involved in political activities. This means they are involved in what we define as proxy culture wars. The short-term outcome is that the progressivists are losing badly. The position of the Neo-Pentecostal churches is so strong that they are attracting a significant number of new entrepreneurs, who mimic the strategies, agendas, and discourse of prospering churches. This internal dynamic is possible only because the proxy culture wars are attractive enough to trigger a local religious sensitivity that is focused on the reassurance of the future and avoiding mishaps (Lindhardt 2014). This process goes hand by hand with the emancipation of the Neo-Pentecostal churches, both economically and strategically, which has enabled them to push forward their independent normative project (Bompani 2017, 2018).

We claim that the advantage of our approach enables us to recognize that Ugandan society is not merely a passive victim of international influence. Quite the opposite is true. There is an authentic struggle between cultural conservatives and progressivists

over Uganda's future. Furthermore, the conservatives are winning, with their high moral authority and influence based on local religious sensitivity.

Our article proceeds as follows: (1.) We discuss vocabulary and (2.) methods. Then, (3.) we describe historical context in connection with the theory of US culture wars and our model of proxy culture wars. After that, (4.) we analyze the Neo-Pentecostal churches and (5.) the NGO narrative of sexual rights along with its implementation into Uganda's reality. We indicate the main reason why Neo-Pentecostals and secular NGOs are in conflict in Uganda. We contend that both ideologies contain a utopian vision that leads them to pursue their goals internationally. (6.) We analyze why Neo-Pentecostal churches are so successful. We argue that Neo-Pentecostals profitably merge Christianity with traditional religious practices. We use examples of churches and NGOs observed during our intensive fieldwork in order to clarify the different levels of involvement within proxy culture wars and Uganda's policy, and to document religious sensitivity attracted to Neo-Pentecostals. We conclude that proxy culture wars are indeed influencing the present development in Uganda. Nevertheless, this international influence is significantly limited by religious sensitivity

## 2. On Vocabulary

As regards the terminology, we use the terms "LGBTQ" and "Neo-Pentecostal" in our study. We decided to use the term "LGBTQ" (lesbian, gay, bisexual, transsexual, and queer) even though conservatives speak about gays or homosexuality only. When we speak about gays or homosexuals, it always in cases where the issue is directly connected with the conservative side of the argument. Additionally, we do not use the more precise lesbian, gay, bisexual, transsexual, transgender, intersex, queer, questioning, 2-spirited, and allies (LGBTTIQQ2SA) concept, which even some queer theory scholars consider too complicated (Finnegan 2002; Alexander and Yescavage 2003).

We also decided to use the term "Neo-Pentecostals" (Born-Again or Balokole) instead of "Pentecostal" only to reflect the changes that took place within the practice and theology of this religious movement: the ability to practice religion in more than one religious society, and the introduction of the Prosperity Gospel (Deacona and Lynchb 2013; Martin 2015; Kalu 2002; Coleman and Hackett 2015; Bompani 2018) (see below).

We also argue for the use of the term "progressivism" instead of liberalism, in order to avoid a misunderstanding with the classical liberalism that is more common in the EU.

## 3. Methods

To investigate roots of the turmoil surrounding the sexual law, we conducted intensive fieldwork during the summer of 2017 in the Kingdom of Buganda (historical part of Uganda), in Kampala and Entebbe.

We interviewed 12 members of the Ugandan Parliament who participated in the passing of the Anti-Homosexuality Bill. Furthermore, we asked them about their views on another law that is taking sexual and moral life under state control. We also asked questions regarding morality, religion, and Uganda's future. In addition, we interviewed six pastors who represented Born-Again movements[5]. We conducted a survey of different types of churches that are members of this movement. Above all, we focused on the African Mountain of Prayer, which seems to be a place where traditional religious and Neo-Pentecostal beliefs merge.

Finally, we interviewed 20 former and current politicians (MPs, ministers), representatives of churches (members of the Inter-Religious Council of Uganda[6]), and secular NGOs. Our sample consisted of Protestant and Neo-Pentecostal pastors, Catholic persons, and Muslim imams. The secular NGOs that we interviewed had been involved in the issue of the LGBTQ minority in Uganda and (or) East Africa, or with the issue of human rights. We focused on the degree of involvement in political activity, patronage, and the international connection of both churches and secular NGOs.

Furthermore, we spoke with Nicholas Opiyo, who serves as a pro-bono human rights lawyer. He was a crucial player in the successful prosecution of the Bill. In addition to that, we invested a large amount of effort in attempting to contact the submitter of the Anti-Homosexuality Bill, David Bahati, and a vocal proponent of the Bill, Simon Lokodo. Unfortunately, neither of them was willing to meet and discuss this issue.

We decided to provide a qualitative study to identify and analyze the international influence and local religious sensitivity. Qualitative research does not aim to be representative. Instead it captures the scope and nature of issues relevant to the chosen field. We used the method of semi-structured interviews for interviewing the participants in our study. We also analyzed texts produced by both NGOs and Neo-Pentecostal churches to support our proxy culture war hypothesis. Analytically, we used thematic analysis to identify themes in data through a process combining inductive and deductive coding (Braun and Clarke 2006). Although the sample of interviews and analyzed texts are relatively small, continuous interviews and text analyses proved their value as new interviews stopped providing new information (Guest et al. 2006). We analyzed religious sensitivity following the theory of Cognitive Science of Religion (Barrett and Lanman 2008; Slingerland and Bulbulia 2011; Szocik 2017; McCauley et al. 2013; Lawson and McCauley 1990; Boyer 2001; Atran 2002). We adopt an evolutionary framework to explain beliefs and practices concerning supernatural entities.

## 4. Culture Wars and Uganda as a Battlefield of Proxy Culture Wars

### 4.1. Pentecostal Influence in Uganda since the 1980s

Uganda in the mid-1980s was, in fact, a fragile state by every standard of this definition (Ferreira 2017), having encountered almost 20 years of either tyranny or armed conflicts. Except for a short period after independence, the people of Uganda had never experienced peace[7]. Yoweri Museveni, the current president, won a long-lasting civil war, but the new regime did not restore peace quickly. The newly formed government lacked the resources to rule over the Ugandan territory successfully. There was nevertheless one actor who substituted the state at a minimum level–churches. Thus began the privileged position of churches in state-wide matters. Later, as a result of government decisions based on international neo-liberal recommendations, the Ugandan government and local councils decided not to provide education and hospital care, and to leave this space for churches (Bompani 2018). Newly legal churches came to Uganda to establish missions, build churches, schools, and hospitals. They probably became the most critical part of many Ugandans' lives at that time.

In 2009, several influential US evangelical pastors led by Scott Lively visited Uganda[8]. During their stay, they delivered several services and public prayers all around the country. At the end of their visit, they even presented their ideas in Parliament. Rev. Lively delivered a speech in which he compared homosexuals to Nazis and pedophiles (Gettleman 2010; Lively and Abrams 2002). Following this, the massive campaign demanding harsh penalties for the LGBTQ minority started. Representatives of Neo-Pentecostal churches lobbied heavily in the constituencies for an anti-homosexual bill, and some of them threatened MPs who were reluctant to do so. The preachers described the LGBTQ community as recruiting children for their sexual needs. Many pastors called for the law to be passed, and have publicly accused politicians of delaying this vital law (Bompani and Valois 2016). The existence of the LGBTQ minority quickly became a burning issue in Ugandan society. The impression was that almost all people called for protection against the members of this sexual minority. They saw LGBTQ as an artificially imported set of practices to threaten African values and identity (Bompani 2017, p. 9).

It has been argued that sexual minorities were never an issue before the intervention by conservative preachers (Kaoma 2009; Vorhölter 2017). This argument is not entirely correct. It only seems to be the case because sexuality was taboo in Ugandan society until the HIV epidemic (Bompani and Brown 2015, p. 114; Bompani 2017)[9]. Silence about sexual minorities does not imply any tolerance directly. However—and more importantly—

Uganda's newly constructed post-colonial national narrativity is connected with Christian conservative values, where National Martyrs[10] play one of the most significant roles (Kizito 2017; Klinken 2020). This means that one of the most important national symbols is of the martyrdom of young males who stood for their Christian faith and refused kings' sexual offers.

Moreover, President Museveni and his administration have brought the discourse of homosexuality in order to align his interests with public opinion, depicting the issue as a foreign threat to Ugandan culture. President Museveni publicly vilified homosexuality more than a decade before the Anti-Homosexuality Bill. It can be recognized as a feature of his pragmatic policy when he often utilizes fear of foreign threats in order to maintain his political control (Sadgrove et al. 2012; Bompani and Valois 2016).

Questions are raised by what motivated the founders of Sexual Minorities in Uganda (SMUG)—an influential sexual rights advocate NGO established five years before Lively's mission in 2004—and what motivated Reverend Scott Lively and his companions to such a severe reaction. Their behavior becomes intelligible only when we employ the concept of proxy culture wars: the present US culture wars being fought worldwide.

*4.2. Culture Wars and Culture Proxy Wars*

We claim that the concept of culture wars enables us to understand what motivates US pastors, religious believers, and members of sexual rights NGOs to intervene in Uganda's culture.

James Hunter introduced the concept of culture wars to the public (Hunter and Wolfe 2006; Hartman 2015; Castle 2019; Hunter 1991). Hunter argues that while the USA has been divided according to a religious idea for its whole history, social changes during the 1950s and 1960s led to a restructuring of religious commitment. The division of the whole society became magnified around cultural values like abortion, women's rights, same-sex marriage, divorce, evolution, religion as part of the school curriculum, and other themes. The New Christian Right (Wilcox 1988, 2018; Coleman and Hackett 2015; Dias 2016) came to represent the cultural conservative wing, while progressivists, represented by organizations like National Organization for Women or American Civil Liberties Union (Hunter 1991; Gilbert 1993; Hunter and Wolfe 2006), fought against this agenda[11].

Despite the long-standing debate over the reality and profundity of culture wars, there is an increasing body of evidence that sufficiently supports the explanatory power of the concept (Castle 2019). We can even expect an acceleration of this cultural conflict (Castle 2019; Murray 2019). According to Castle's research, the fundamental factor of this deep polarization that can mark each position in the culture war is religion in both senses: as an affiliation with religious tradition and as religious commitment (Castle 2019).

We propose the neologism proxy culture wars for the cultural development in Uganda in order to emphasize the analogy with numerous locally isolated international conflicts fought during the Cold War. We argue for the application of the concept of proxy wars with culture wars because—as we discuss–US culture wars are being fought internationally.

Several factors motivate progressivists and conservatives to clash abroad. The first important issue connected to our argument is that despite the deep division around cultural values, the structure of the values and strategies of both sides are similar. Both sides view their values as universal without any exceptions. Both sides of the conflict try to shift the public discourse according to their values and beliefs.

Hartman describes conservatives' cultural standards as "normative America" (Hartman 2015, p. 3). By this he means the whole cluster of beliefs and assumptions that characterized the USA during the 1950s. The inner structure of their values is not only normative but universal. They are dedicated to a universal value of transcendent authority, accompanied by the idea of cultural essentialism (Fumanti 2017, pp. 4–5). They commit their lives to ongoing proselytism. Their fight is ontologically fatal because of its substantial apocalyptic understanding of human history (Haidt 2012). They are devoted to protecting society from threats of "godless" reforms that—in their eyes—bring catastrophic conse-

quences. All social reforms according to them are based on ignorance (Haidt 2012; Boyer and Petersen 2018). By ignorance, they mean a lack of belief in the truth of the Gospel.

On the other hand, the progressivists are devoted to the universal ideal of expanding human rights (Hunter and Wolfe 2006, p. 2; Fumanti 2017). They also fight suffering throughout the globe, but through the lens of "social justice," "identity group politics," and "intersectionalism" (Murray 2019, p. 12; Rectenwald 2019). Their ideology is mostly derived from the left-wing of the political spectrum, and post-modernism (Allen and Flynn 2018; Fernández-Aballí 2016; Murray 2019; Rectenwald 2019). It follows that they interpret the reality as constructed all-the-way-down, that reason and power are the same (Lyotard 1984; Foucault 1978), all the widespread oppression lies in the cultural West, and the highest virtue lies in social activism (Hicks 2011; Hartman 2015; Rectenwald 2019).

The inner utopian structure of both ideologies fuels the urgency of their struggle (conservative and progressive activists). Both sides of the conflict analyze the present situation as deficient. Both define a goal that should be reached via radical changes and intervention in the present policy.

The reason that brings conservatives to Uganda is the fact, taken from the situation in the USA, that the religiously oriented Protestant majority is now more and more marginalized (Castle 2019, p. 654). From the conservative point of view, there is a real threat that culture wars are going to be a lost cause on US soil (Castle 2019; Murray 2019).

The reason that brings progressivists to Uganda is the fact that LGBTQ rights are part of human rights (Ayoub 2017; Ayoub and Page 2020; Castle 2019; Hartman 2015; Murray 2019; Rectenwald 2019), and, therefore, an inevitable part of humanitarian aid.

The clash over the Anti-Homosexuality Bill of 2009 made the conflict between sexual rights NGOs (progressivists) and Neo-Pentecostals (conservatives) visible (Kaoma 2009, 2012; Watt 2009). To understand proxy culture wars better, we need to look closer to both sides of the narrative and their presence in Uganda.

## 5. Neo-Pentecostals and Their Narrative

Neo-Pentecostals follow the original doctrine of Pentecostalism, which is a part of the American spiritual movement that broke up in the so-called Bible Belt in the South of the USA in the 19th Century. The literary belief in the Bible, speaking in tongues and spirit possession are the most characteristics common to all Pentecostal churches.

What changed Pentecostalism to Neo-Pentecostalism was the dedication of their lives in more than one religious society. Since the 1960s, this practice has been introduced. Anyone can stay in her or his congregation and communicate her or his new spiritual experience to other members of the church.

Another thing that became typical for Neo-Pentecostalism was the introduction of Prosperity Gospel (Deacona and Lynchb 2013; Martin 2015; Kalu 2002; Coleman and Hackett 2015; Bompani 2018)[12]. The core message of the Prosperity Gospel is that all Christians are inevitably meant to be wealthy, because Christianity means wealth. Poverty is a significant indicator of personal sins or spiritual problems, curses, or obstacles (Deacona and Lynchb 2013, pp. 109–10). Everyone is called upon to break this kind of curse through donation, visualization, and positive confession.

The culture wars front during the 1990s mostly concerned the issue of gay rights (Hartman 2015; Murray 2019; Ayoub 2017; Castle 2019). Christian conservatives believe that homosexuality has nothing to do with a natural disposition. They believe that it is learned behavior. A person needs to choose to become gay (Murray 2019). Theologically speaking, they link homosexuality with sodomy. That means there is pure evil in homosexuality, connected to God's enemy. They also do not distinguish between pederasty and homosexuality (Tamale 2013).

When Lively talks about his personal experience, he gives an example of a gay person that molested a four-year-old boy, has never been punished, and is an active member of a gay-tolerant church[13].

This view can be even radicalized. The group represented by Reverend Lively believed in a LGBTQ conspiracy. Lively expresses his belief in an LGBTQ conspiracy in his book *The Pink Swastika* (Lively and Abrams 2002)[14]. He and Abrams construct a lineage between the fight over the character of present culture in the world and the Second World War. Lively even holds members of the LGBTQ community responsible for the Holocaust (Lively and Abrams 2002).

When Reverend Lively was, in 2012, charged by The Center for Constitutional Rights with "Crimes Against Humanity of Persecution" as a consequence of his activity in Uganda, this merely reinforced his belief in an international LGBTQ conspiracy (Lively 2015)[15]. Moreover, both Lively and The American Family Association[16] (AFA), who supported him during the lawsuit, have been listed by a human rights NGO—the Southern Poverty Law Center[17]—as a far-right extremist and AFA as an anti-LGBTQ hate group.

There are several texts spread over Kampala documenting the influence of Christian conservative values and efforts to connect US internal policy with the African environment. Particularly interesting is a booklet that we noticed in the waiting room of The Victory Christian Centre Church (Ndeeba, Kampala). The church was founded by Dr. Joseph Serwada, one of the Neo-Pentecostal representatives in the Inter-Religious Council of Uganda[18].

The story in the booklet was not only about conversion to Christianity as usual, but also about political conversion. This fascinating piece depicted President Trump as a Born-Again protector of Christianity and conservative values around the world (Trip and Luo 2016). The main character was a young girl that followed liberal values and suffered from the consequences. Then, she found Jesus and conservative values as represented by President Trump. It changed her whole life in a better way. Her health improved, as well as her financial situation. She was saved in both an earthly and spiritual sense.

Although the message concerning conservative values rooted in divine sources was clear, the language itself was puzzling, taking into account the reality of Uganda. When we spoke with members of different Neo-Pentecostal communities, they displayed the only a vague or even no understanding of US internal politics. Nevertheless, they confidently saw president Trump as a popular figure and their moral defender[19].

The Neo-Pentecostal worldview is apocalyptic, generally speaking. The powers of God and evil are real. The cultural West is decadent and morally corrupted. Sexuality—a controversial issue for Christianity from its institutionalization—makes moral corruption vivid. The LGBTQ agenda is the epitome of evil because, according to Neo-Pentecostals, its target is the family.

An example documenting both the apocalyptic message and the influence of Culture Wars is taken from the webpage of Africa Prayer Mountain for All Nations. The mountain is located about 14 km (about 8.6 miles) from the center of Kampala. The founder of the Mountain of Prayer is Rev. Dr. John W. Mulinde[20]. The second co-founder is pastor Mark Daniel, who lives in Orlando, Florida. Rev. Mulinde was also one of those pastors that backed the Anti-Homosexuality Bill (Kaoma 2012).

Mulinde based his legitimacy on his vision of God calling for humanity to be prepared for his return[21]. Therefore, the movement has its apocalyptic dimension when the world is depicted as corrupted, emphasized by the language of spiritual struggle. Mulinde specifically mentions the corrupted lifestyle of modern times: "evil passions, perversions, and obsessions that made them [people] crave for what was unnatural."[22] He is obviously on the side of the cultural conservatives when he mentions progressivists: "In other cases, the enemy has turned it into free liberalism, which allows people to gratify all their human desires, in total disregard of God or His Word."[23] In other words, political and cultural progressivism is a project of the devil himself. The whole webpage, especially those passages against cultural and political progressivists, is heavily influenced by US religious conservativism, not easily comprehensible to any typical member of the Africa Prayer Mountain church. When we asked members of this community about political and cultural

progressivism, they did not understand what we meant. However, when we reformulated Mulinde's entire proclamation into moral language that concerned conservative and progressive values, they were unequivocal supporters of conservative values.

Last, but not least, is the financial support for Neo-Pentecostal churches that comes from private and government sources. The financial support for Neo-Pentecostal churches goes hand in hand with the change of US government agenda for international aid to combat the HIV/AIDS epidemic (Hartman 2015, p. 158; Gusman 2009; Padamsee 2020; Boonstra 2003, p. 2; Bompani 2018, p. 307), which, beginning in 2004, redirected all HIV aid to "morally informed campaigns." This means to conservative Christian values-based organizations (Bompani and Valois 2016, p. 4; Cooper 2015, p. 55). What followed was President's Museveni newly installed strategy to change the original HIV/AIDS policy (the so-called "ABC mode": abstain, be faithful, use condoms) into "abstinent" behavior[24] (Boyd 2015). Since then, Neo-Pentecostal churches have qualified for US financial aid, as well as for the influence of US conservative Christian recommendations.

Fox Odoi, a former MP, documents the connection between local Neo-Pentecostal preachers and their US supporters. He said: "The Neo-Pentecostal movement has a strong connection with the Americans. They lent them money to build churches and sent them used megaphones to shout prayers on the streets. These Neo-Pentecostals from the States are trying to export what they have been trying to achieve in the USA."

In other words, it is a strategy of Neo-Pentecostals to use texts, as well as political and financial support, to protect Uganda from what they considered to be evil of an apocalyptic dimension.

## 6. Progressivists, LGBTQ Rights NGOs, and Their Narrative

The issue of sexuality and rights of sexual minorities and their political struggle to be recognized as a group protected by the Constitution goes back to the 1960s in the USA. At the very beginning it was a movement of gay men, which became radicalized during the 1980s due to the struggle with HIV/AIDS (Eisenbach 2007; Adam 1987; Aldrich 2010). The political change happened during the 1990s almost overnight.

On the international level, sexual rights were accepted as the international norm at the United Nations Conference on Population and Development in Cairo in 1994 (Picq and Thiel 2015). It subsequently led to the Brazilian Resolution on human rights and sexual orientation that was presented to the ECOSOC in 2003 (Girard 2007; Picq and Thiel 2015; Otto 2017). Later, the Declaration of Montreal (2006) summed up the protection of LGBTQ human rights, and Yogyakarta Principles (2006) and their supplement (2017) demonstrate the inclusion of LGBTQ and human rights.

The view of LGBTQ rights rapidly changed in the majority of Western countries. Although, at the end of the 1980s, the majority of culture reflected the gay community as immoral, the majority of the population of the USA and the EU now see gay marriage as legitimate. This has happened mainly thanks to well-organized LGBTQ groups that cooperate on the international level and is represented mostly—but not exclusively—by organizations like the International Lesbian and Gay Association (ILGA), the International Gay and Lesbian Human Rights Commission (IGLHRC), Human Rights Watch, the Sexual Rights Initiative and other initiatives based on a national or regional level (Picq and Thiel 2015, p. 2).

Progressivists focus on several values that are fiercely contended issues between them and cultural conservatives. One of the crucial issues was gay rights, which quickly transformed into LGBTQ rights. Progressivists support the idea that sexual minorities' rights are part of human rights and, as such, need to be supported worldwide (Picq and Thiel 2015, p. 14). Moreover, the recognition of sexual rights for minorities becomes one of the markers for progress in the social justice agenda.

Despite the fact of a well-organized cooperation, members of the LGBTQ community are far from unified (Picq and Thiel 2015, p. 4). Nevertheless, we can speak about the general framework of LGBTQ discourse. They follow the agenda of cultural liberals, who

are prominent in the field of "gender studies," "women's studies," and other "minority studies" in academia. They have predominantly adopted Queer theory, which received its first impulse from post-Marxist ideology and post-modern philosophy. Authors like Laclau and Mouffe, McIntosh, and Butler (McIntosh 1988; Butler 1990; Laclau and Mouffe 2001) take as a basis the philosophy of Michael Foucault and Gilles Deleuze in order to reformulate the Marxist class-struggle into a struggle of oppressed minorities against cultural hegemony, sexual or gender minorities included. According to them, society is not a compound system of relations, values, and cooperation but a totality of diffused power (Murray 2019; Hartman 2015). There are cultural hegemons (white, heterosexual, cisgender, men), against whom are many groups that need to fight for their identities to demand social justice.

Queer theory means the radicalization of the LGBTQ issue in the sense of opposing the idea of a fixed sexual identity in favor of a fluid performative character of sex or gender. According to Queer theory, the ideal of fixed sexual identities are a product of patriarchal oppression and the normative character of the modern state (Bosia 2015; Browne and Nash 2014). The basic anti-normative and anti-ontological character of Queer theory rejects any fixed social reality (Picq and Thiel 2015, p. 13; Bosia 2015). As such, Queer theory has a utopian character, and activism is a practice for this ideology.

Members of sexual and gender minority rights NGOs feel called upon to fight social injustice and inequality over the globe. LGBTQ organizations work internationally (Ciszek 2017; Picq and Thiel 2015; Browne and Nash 2014; Murray 2019). They focus on public relations in order to cooperate strategically to develop a new discourse, subvert traditional cultural identities, representations, and imagination (Ciszek 2017, p. 703). According to Ciszek, LGBTQ activists have built up an international social media infrastructure that has more than 1.2 million followers (Ciszek 2017, p. 707). The U.S. based parent organizations manage strategies, sources, and organize cooperation on the international level (Ciszek 2017, p. 708) to achieve outcomes.

There are two factors that concern Uganda and the whole of Sub-Saharan Africa. First of all, LGBTQ rights are recognized as an inevitable part of human rights, and, as such, deserve support. Therefore, they are part of humanitarian aid to the countries of the Third World. LGBTQ rights are not only monitored by LGBTQ organizations but have become a legitimate part of a significant marker of the human rights situation and progress around the world.

The second factor is the highly sophisticated international cooperation of LGBTQ activists, who are internationally represented by organizations like International Lesbian, Gay, Bisexual, Trans And Intersex Association[25] (IGLA), the International Gay and Lesbian Human Rights Commission (IGLHRC), Human Rights Watch, the Sexual Rights Initiative, the Coalition of African Lesbians (Picq and Thiel 2015; Murray 2019; Ciszek 2017; Otto 2017), and Transgender Equality Uganda (Bosia 2015).

The level of international outrage immediately after the Anti-Homosexuality Bill was passed demonstrates the highly successful strategy of the LGBTQ lobby and human rights organizations. Various kinds of Western media referred to the situation in Uganda[26]. Even later, the media has attempted to keep up coverage of the situation in Uganda.

Moreover, sexual rights NGOs can react rather swiftly in case of a threat. Nicholas Opiyo (lawyer and human rights attorney) remembered one occasion when the LGBTQ community gathered in a hotel in Kampala. They were almost instantly asked to leave the hotel conference hall, and police started to arrest people there. In less than an hour, several western ambassadors came to the hotel saying they were invited to the gathering, and the police officers were forced to back out. According to Opiyo, even the daughter of late US president J. F. Kennedy—Caroline—called upon President Museveni to not intervene in this matter.

In addition, the member of Sexual Minorities Uganda (SMUG)[27] Frank Mugisha was awarded the Robert F. Kennedy Human Rights Award (2011), which we can interpret as a sign of the promise of international human rights organizations and political activists not

to abandon the LGBTQ community in Uganda. The award is issued to promoters of human rights, sexual rights, and gender identities. A specific feature of this award is long-time strategic cooperation with laureates and associated organizations.

It is not only Neo-Pentecostal churches that use language which seems to be influenced by the milieu of US cultural wars. We can mention the Sexual Minorities Uganda (SMUG) NGO as a compelling example documenting the influence of progressivist ideas. The organization itself was established in 2004 in order to support the LGBTQ minority rights and also coordinate a further 18 LGBTQ organizations in Uganda. The ultimate goal for SMUG according to its webpage is to "ensure that all Ugandans are equally respected and valued no matter their sexual orientation, or gender identity or expression." SMUG's activities are also internationally coordinated, with SMUG involved in all kinds of international commemorations important for the LGBTQ minority, such as the celebration of Transgender Day Of Remembrance–TDOR–and others.

Queer theory heavily influences the language used on its website. In contrast to the Neo-Pentecostals, who always use terms like "gay" or "homosexuals," the SMUG website has progressively moved from the term LGB to LGBTQ+[28].

In contrast with the ability of international cooperation is the relative failure to change the public discourse in Uganda. The language of the website is practically incomprehensible to anyone in present day Uganda, except for highly educated individuals. Our respondents usually displayed disbelief or distress when we tried to explain Queer theory to them during our interviews. When we reformulated Queer theory into a story, their response was on a scale from reserved acceptance of the idea of personal freedom and justice (employees of secular NGOs), genuine disbelief (members of traditional Christian denominations), to pure distress (members of Born-Again sects)[29].

One of our respondents said: "Well, God forbid I don't want to think of it. I pray that none of my kids fall for that because it is disgusting. It is against our culture and religion. There is no justification for that."

***

Sexuality was taboo in Uganda, and it became a public issue only at the time of the HIV epidemic (Bompani and Brown 2015). On the other hand, it is one of the vital issues for both conservatives and progressivists in the new round of culture wars (Castle 2019; Hartman 2015). For conservative Christianity, homosexuality (LGBTIQ) is (next to feminism) another crucial attempt by progressivists to destroy the traditional family. For progressivists, on the other hand, it represents a vital part of the process of expanding human rights (Hartman 2015, p. 156; Castle 2019, p. 654; Miller et al. 2017; Lewis et al. 2021).

It is plausible to claim that for anyone who comes to visit Uganda, the human rights of sexual minorities would not be the first thing to cross her or his mind. Overwhelming poverty, lack of access to clean water, and child mortality to mention just a few seem to be the most striking issues. Nevertheless, nothing has brought outrage to the West—even the internal politics of president Museveni—except the Anti-Homosexuality Bill (Bompani and Valois 2016, p. 11).

Our respondents frequently displayed disbelief and curiosity about why the West is concerned about the problem of a sexual minority, while remaining inert when the fundamental values of democracy are at stake? Neo-Pentecostals connected to the New Christian Right and NGOs connected to the LGBTIQ organizations clashed over their values in Uganda.

By analyzing the theory of culture wars, and the Neo-Pentecostal and sexual rights NGOs narratives, we present a clear understanding of why both sides of US cultural wars have clashed in Uganda. We claim that there is a utopian vision as part of both ideologies that force them to pursue their goals internationally—Uganda's cultural environment responds with high sensitivity to both ideologies that are associated with different values.

In the next section, we analyze religious sensitivity, which is a crucial phenomenon for understanding why cultural conservatives are convincingly winning the proxy culture wars in present day Uganda.

### 7. The Success of Neo-Pentecostalism: Religious Sensitivity Gives Proxy Culture Wars Correct Proportions

We contend that an understanding the concept of proxy cultural wars enables us to comprehend Uganda's cultural development, which correlates with the intensifying culture wars between cultural conservatives and progressivists in the USA.

Nevertheless, proxy culture wars alone are not sufficient for a full understanding of Uganda's reality. The existence of the proxy culture wars can explain the international outrage, media interest, motivation of missionaries, NGO workers, and financial support, but it does not explain the attractiveness of the Born-Again movement for the rising number of believers.

We assert that Neo-Pentecostals are successful because they effectively merge their theology with local religious sensitivity. The concept of religious sensitivity is the second feature that gives proxy cultural wars appropriate proportions and helps us to understand why the conservatives are winning.

We understand religious sensitivity as an evolved ability of the human mind to intuitively follow allegedly supernatural visions (in a broad sense), shaped according to local tradition and cultural history[30].

Moreover, we have decided to speak about religious sensitivity rather than religious belief[31] because the study shows that our traditional understanding of religions as a set of coherent propositions is over-intellectualized (Slone 2007; Peedu 2016; Shariff et al. 2016; Barrett 2008; Reddish et al. 2016). Religion/belief is mostly used as a kind of practical instinct that is involved in all parts of life. All the interlocutors were puzzled when we asked them about differences between the profane and the sacred, between public life, politics, family, and religion. We changed our questions into a number of stories to be able to formulate precisely the position of the supernatural in everyday experience. One interlocutor labeled our questions as strange stories.

In addition, our story, when we described a person who questioned the existence of God, was described by our informants as even more peculiar. One of our interlocutors commented: "What a strange story. Of course God exists. Who would ask such a silly question? It is like the weather. You never ask if you believe in the weather? Why would you? The weather just is!" This example explains the number of believers in Uganda. The existence of the supernatural is accepted intuitively, operating beyond any doubt.

However, why Neo-Pentecostalism, and not Islam or Roman Catholicism? The source of Neo-Pentecostals' popularity lies in their ability to merge with traditional religious practice (Lindhardt 2014). What makes Neo-Pentecostals particularly attractive is their minimal stress on theology and maximum of practice. Moreover, their practice is heavily based on the informal religious tendency[32] to secure future prosperity, and to avoid mishaps (Boyer 2020a, p. 1). This kind of practice is present everywhere, and is exceptionally strong where a modern state does not exist or where the state is fragile, like in Uganda.

Neo-Pentecostals also emphasize the informal religious tendency more than any doctrine. Neo-Pentecostal practice, with its minimalistic ritual, strongly resembles the practice of Kiganda—the traditional way of worshipping[33]—that consisted mainly of a set of practices to treat the spirits of ancestors and other spirits to avoid misfortune and a fixed future (Ray 1991).

When we compare a ritual in the semi-cloistered shrine in Bulenga with a service in the New Miracle Church[34] (Kazo, Kampala), the very core of the ritual remains identical. The worshippers are offered the prospect of wealth, health, and a suitable sexual partner. When we analyze the collected empirical material from seven churches, the results remain the same. Some of them are the most significant mega-churches like the Africa Prayer Mountain for All Nations, Watoko Church, the Victory Christian Centre Church (Ndeeba,

Kampala), Miracle Center Cathedral (Mengo, Kampala), Synagogue Church of All Nations, and some of them are relatively small but quickly growing, like the True Vine Ministry Church (Kiwafu, Entebbe), and the aforementioned New Miracle Church (Kazo, Kampala).

There is enormous competition between different churches and shrines. It is always the same phenomena that make those places credible, namely healing miracles and casting out of devilish spirits that possess believers and cause misfortune and illness. Nevertheless, people are not so gullible as to believe everything (Mercier 2020), because any mistake can be fatal from the evolutionary perspective (Sperber et al. 2010; Norenzayan et al. 2016; Mercier and Sperber 2017; Harris et al. 2018). As a result, people carefully observe significant signals from different churches, and their epistemic vigilance is always present (Sperber et al. 2010; Norenzayan et al. 2016; Mercier and Sperber 2017; Harris et al. 2018). There are ongoing rumors about pastors who cheat and provide some cheap tricks only. In the age of the Internet, there are many circulating videos accusing different pastors of providing effects only[35], instead of genuine miracles. The fear of being subject to a scam is present everywhere, because resources are scarce.

Contrary to the existing rumors of fake miracles, the numbers of congregations are rising. The explanation lies in the pastors' behavior. Pastors use expensive signaling (Bliege Bird and Power 2015; Boyer 2018; Mercier 2020; Hall and Gonzales 2017; Mercier 2017; Park et al. 2017) to present themselves as them credible and to show their devotion to the Born-Again movement. Because they follow a traditional pattern of religious sensitivity, they are sometimes tempted to give another signal of credibility and status, which is the possession of multiple wives despite Christian monogamy ("Father Lokodo Threatens to Suspend Pr Bujjingo–Sqoop–Its Deep" n.d.).

Traditional churches, such as the Church of Uganda, and Roman Catholics criticize pastors for displaying a luxury lifestyle among their impoverished followers. It seems to be counterintuitive to support wealthy pastors when people are struggling to support their large families. The reason why this strategy is successful is that it makes the global economy intelligible. Experiments show that our understanding of economic processes is limited. It has its roots in simple exchange in small scale societies (Boyer and Petersen 2018; Boyer 2018). Contrary to the evolved folk-economic beliefs, contemporary wealth on a global scale is produced as a result of an aggregation of many simple processes that go beyond our understanding (Boyer 2018, pp. 177–86). Therefore, it triggers an intuition of the supernatural origin of wealth. Rich pastors signal to believers that they can make them rich in the same way. This process—associated with the Prosperity Gospel—works not only in Uganda (Martin 2015; Deacona and Lynchb 2013; Bompani 2018; Coleman and Hackett 2015; Niemandt 2017).

The second powerful signal that Neo-Pentecostals issue, which is also plausible in connection with traditional religious sensitivity, is miraculous healing. It also combines folk-medicine understanding with success bias (Arocha et al. 2017; Whyte 1997; Mercier 2020). Illness is intuitively understood as an attack from outside by harmful spirits (Whyte 1997). Because supernatural forces outside of the body cause all illnesses, it is only natural to heal them supernaturally. We witnessed several miraculous healings based on exorcism. The number of healed people is a vital signal that attracts people to the church.

What helps besides religious sensitivity is success bias. Any improvement that happens during or after a religious service is recognized as miraculous help. In this case believers spread information about the miraculous abilities of a reverend. If the health of believers does not improve, they are occupied with maintaining an intensive search for another kind of help. Thus only positive information is spread (Boudry et al. 2014).

It is not a person but a detached divinity (Boyer 2020b) that guarantees the credibility of the church or shrine and confers prestige and higher moral standard upon a person. Even our Muslim respondents recognize the power of different churches and shrines[36].

The natural tendency to understand signs of divinity gives the Neo-Pentecostals a moral authority, which is plausible within the tradition of the Big Man (Lindhardt 2014, p. 15; Lauterbach 2017). It also naturally associates them with politicians. This explains

why the Neo-Pentecostals are significantly more connected to politicians than any other religious group in Uganda (Bompani and Valois 2016; Bompani 2017; Bompani and Brown 2015). Neo-Pentecostal reverends are mostly welcomed for their moral, spiritual, and economic influence. They are recognized as people with a spiritual power, a higher moral authority (Bompani 2018), and good connections with Western countries.

Our findings also support the idea that anyone who is connected with the "Born-Again" church increases her or his trustworthiness and impression of incorruptibility (McKay and Whitehouse 2015; Shariff et al. 2014, 2016; Gervais et al. 2017). Almost every MP that we interviewed admitted that she or he was at least in touch with one or another Neo-Pentecostalian pastor, if not directly part of the Born-Again movement.

The predilection of politicians for Neo-Pentecostal churches is not merely a matter of pragmatic exploitation. Politicians also associate themselves with a particular church based on "divinity", the presence of which guarantees the credibility of this church. Politicians see signals and realize intuitively what a significant role these churches play in their constituencies. The number of Neo-Pentecostal MPs has also risen steadily in the past years, especially among youth representatives. They seek supernatural and social support. It is an extensively shared strategy (not only) for members of parliament to seek divine help to ensure their position. Every Member of Parliament we spoke to admitted that she or he consulted Neo-Pentecostal pastors in order to seek support.

In summary, Neo-Pentecostals are popular because they offer stability and fixed values that are guaranteed by divinity. They are also able to render intelligible complex systems of open economic and healing processes on an intuitive level. Altogether, this produces a success bias (Henrich 2015, p. 49), which generates a rapid increase in the number of believers.

When we compare this with the cluster of beliefs that progressivists are offering, there is no doubt as to why proxy culture wars favor the side of cultural conservativism. Progressivists and Queer theory offer the idea of a fluid reality, which means more instability. To seek instability is not intuitive at all, and progressivists are not able to offer any guarantees of reliability.

## 8. Conclusions

Our study sought to investigate what has caused turmoil concerning the sexual laws in present day Uganda. In doing so, we analyzed texts and sermons produced by both sexual rights NGOs and Neo-Pentecostal churches. We interviewed 12 members of the Ugandan parliament, as well as 20 former and current politicians and representatives of churches.

Our findings support the original hypothesis that the triggering moment for unrest in relation to sexual law is proxy culture wars (the US culture wars fight internationally). Nevertheless, we discovered that the effect of the proxy culture wars is significantly limited by religious sensitivity.

We contend that the advantage of our approach enables us to recognize that Ugandan society is not just a passive victim of international influence. Quite the opposite is true. There is an authentic struggle between cultural conservatives and progressivists over Uganda's future. Furthermore, conservatives are winning, with their high moral authority and influence based on local religious sensitivity.

By religious sensitivity we mean those abilities of the human mind that have evolved during cultural evolution, and which are attracted to seek supernatural reassurance of future avoidance of mishaps, as well as what Boyer characterizes as wild tradition (Boyer 2020a).

In Uganda, as in many other Sub-Saharan countries, there is a strong connection between religion and politics. On one hand, this connection can be minimalistic: for example, the incumbent pays a prearranged amount of money in his church for re-election, in order to be sure that the pastor will pray for him on an election day. On the other, it can escalate to systematic pressure to push religious agenda into laws. The latter, far more

radical form of connection already happened in Uganda in 2009, and many circumstances indicate that it can happen again.

**Author Contributions:** Conceptualization, M.P. and T.T.; methodology, M.P.; software, M.P. and T.T.; validation, M.P. and T.T.; formal analysis, M.P. and T.T.; investigation, M.P. and T.T.; resources, M.P. and T.T.; data curation, M.P. and T.T.; writing—original draft preparation, M.P. and T.T.; writing—review and editing, M.P. and T.T.; visualization, T.T.; supervision, M.P.; project administration, M.P. and T.T.; funding acquisition, M.P. Both authors have read and agreed to the published version of the manuscript.

**Funding:** This research was funded by Philosophical Faculty (UHK), Specific Research 2017 and The APC was funded by Philosophical Faculty (UHK).

**Institutional Review Board Statement:** The study was conducted according to the guidelines of the Declaration of Helsinki, and approved by the Institutional Review Board of University of Hradec Kralove (UHK) (Specific Research Board, Philosophical Faculty (UHK); approved 8 March 2017).

**Informed Consent Statement:** Informed consent was obtained from all subjects involved in the study.

**Data Availability Statement:** The data presented in this study are available on request from the corresponding author.

**Conflicts of Interest:** The authors declare no conflict of interest.

## Notes

1　More information about this topic is presented in the documentary God Loves Uganda (2013).

2　Illegal since 1950, but the law came into practice more systematically after 2009.

3　Moreover, it represents the Ugandan people as passive victims of international influence. Taken from our fieldwork, nothing is farther from reality than the idea of local people as objects of foreign policy.

4　Alongside the radical views inspired by neocolonial theory on the one hand and by gender studies on the other, Alava and Ssentogo point out that there is a tendency of de-politicization within religious societies in present day Uganda (Alava and Ssentongo 2016). Moreover, Bompani coherently claims that Neo-Pentecostal churches are also pursuing an independent normative project of reshaping Uganda's public space, morality, and public sexual behavior (Bompani 2017, pp. 11–12).

5　We interviewed pastors in the African Mountain of Prayer, the True Vine Ministry Church in Kiwafu, Entebbe, the New Miracle Church in Kazo, the Victory Christian Centre Church in Ndeeba, Kampala, the Miracle Center Cathedral, Mengo, Kampala, and the Synagogue Church of All Nations, Kampala.

6　https://ircu.or.ug/ (accessed on 1 July 2021).

7　Even though Milton Obote's regime fell, a new threat known as the Lord's Resistance Army (LRA) led by Joseph Kony rose up in northern parts of Uganda (Day 2019).

8　Anglicans (Church of Uganda) and Roman Catholics constitute the majority of the Ugandan population. However, their reflection of the fast-developing situation is beyond the scope of this article. For more details, see Hansen and Twaddle, and also Chapman, who offer a well-informed analysis. (Hansen and Twaddle 1995; Chapman 2018).

9　By referring to sexuality as taboo, we mean that sexuality was not part of polite conversation. However, sexuality has been part of legislation dating back to the colonial era. (Peterson 2012).

10　The story commemorates the last pre-colonial Buganda king, Mwanga II, who executed 45 young pages because they rejected his sexual offers after they converted to Christianity.

11　The situation in the European Union is slightly different. It is insufficiently studied, but we can assume that the clash over the values resembles the situation in the USA, with some EU specifics that are given historically and culturally. The critical differences between the US and the EU are in the significance of religion. Generally speaking, religion is not that influential in the EU. The influence of churches is stronger in southern countries (Italy, Spain, Portugal, Greece), and is weaker or absent in the northern part of the EU (Scandinavia, Germany, Czech Republic, Baltic states).
It is debatable as to whether cultural wars developed separately in the EU or a result of US influence. There is a high probability that it concerns several different projects (Paternotte and Kuhar 2018, p. 14). Nevertheless, it is evident that a clash over values—less intense than in the US—is present (Paternotte and Kuhar 2018; Ozzano and Giorgi 2015). There is an intensifying number of public debates relating to political development in the EU, such as the veil affair in France (2004–2011), same-sex marriage law across the EU (from Belgium in 2003 to Austria in 2019) or the migration crises (2015) (Paternotte and Kuhar 2018; Ozzano and Giorgi 2015). From another point of view, the influence of the EU is present in Africa when the EU mostly supports progressivist NGOs (Crawford 2017).

Moreover, Neo-Pentecostal pastors like Scott Lively despise EU policy as progressivist "all the way down" (Lively 2015).

12   The Prosperity Gospel was introduced by American evangelists such as Kenneth Hagin and Kenneth Copeland during the 1960s and 1970s (Deacona and Lynchb 2013, p. 110).

13   https://www.youtube.com/watch?v=e9F9k4guN3M (accessed on 1 July 2021).

14   He has recently been working on the fifth edition http://www.scottlively.net/the-pink-swastika/ (accessed on 1 July 2021).

15   He was charged on behalf of SMUG. The lawsuit received substantial publicity (Scott Lively Says Being Gay Is 'Worse than Mass Murder' as Trial Moves Forward 2014; Goodstein 2012; Sexual Minorities Uganda v. Scott Lively | Center for Constitutional Rights 2016; Halper 2012). Nevertheless, the case was dismissed in 2017 due to a lack of jurisdiction (Johnson 2017).

16   http://www.afa.net/ (accessed on 1 July 2021).

17   https://www.splcenter.org/ (accessed on 1 July 2021).

18   ircu.or.ug (accessed on 1 July 2021).

19   According to our survey, members of the Born-Again community view President Trump positively, while members of traditional Christian denominations are more reserved in their judgement. Muslims mostly see President Trump as a threat. President Trump remains popular in Africa in general (Why Donald Trump Is Popular in Africa 2018).

20   The original description of the prophetic dream that commanded rev. Mulinde to found the church says that it happened in 2015. The website of the World Trumpet Mission says that it initially occurred in 1988: http://www.worldtrumpet.com/the-testimony (accessed on 1 July 2021).

21   https://worldtrumpet.wixsite.com/world-trumpet/the-testimony (accessed on 1 July 2021).

22   https://worldtrumpet.wixsite.com/world-trumpet/the-testimony (accessed on 1 July 2021).

23   https://worldtrumpet.wixsite.com/world-trumpet/the-testimony (accessed on 1 July 2021).

24   Behind Museveni's decision is probably the influence of his wife Janet, who is a declared Born-Again believer (Gusman 2009, p. 71).

25   https://ilga.org/ (accessed on 1 July 2021).

26   For instance, John Oliver covered the Ugandan situation in at least three episodes of his show (https://youtu.be/G2W41pvvZs0 accessed on 1 July 2021).

27   The list of the prominent LGBTQ members published by Rolling Stone magazine was practically a list of SMUG members.

28   https://sexualminoritiesuganda.com/call-to-action-by-ugandas-lgbtqi-community-statement/ (accessed on 1 July 2021).

29   Other NGOs involved in Uganda proxy culture wars like Wellspring Advisors, Human Rights Awareness and Promotion Forum (HRAPF), DefendDefenders, The Uhuru Institute, or Chapter Four are mostly devoted to an agenda including racial, gender, and economic justice. They are clearly progressivists. Nevertheless, they only display a mild influence of the queer philosophy.

30   A lengthy debate is ongoing over (Neo)-Pentecostalism, concerning whether (or in what senses) Pentecostal churches draw more upon cultural continuity or cultural discontinuity. Matthew Engelke and Birgit Meyer, for example, argue for its novelty (Meyer 2004, 2021). On the other hand, Nimi Wariboko (Wariboko and Afolayan 2020) and Paul Gifford (Gifford 2016) accentuate its cultural continuity or its African-ness. We believe that a study of Pentecostalism as a cluster of semi-independent processes following the Cognitive Study of Religion will be more helpful. However, we are aware of the fact that our religious sensitivity hypothesis needs to be supported by further studies in the future.

31   As recent development among CSR shows, religious belief is a rather slippery concept (Peedu 2016; Purzycki et al. 2012; Barrett 2017; Szocik 2017; Boyer 2018; Purzycki et al. 2018; McCauley 2013). It is a relatively new concept that seems to be relevant for a historical study of the Reformation in the 16th century, when both sides—Protestants and Catholics—drove themselves to a position of needing to formulate their theology explicitly.

32   or "wild tradition" as Boyer puts it (Boyer 2020a, p. 1).

33   Literally "the way things are done by Baganda".

34   https://www.facebook.com/New-Miracle-Center-Church-Kazo-475400949229529/ (accessed on 1 July 2021) The founder is Reverend Harriet Mugerwa.

35   One example: https://storyteld.net/10-crazy-things-men-god-followers-last-decade/ (accessed on 1 July 2021).

36   They denied entering any of the Neo-Pentecostal churches, with the argument that they would not risk possession by evil spirits.

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
