# Peer review of "The Limiting of the Impact of Proxy Culture Wars by Religious Sensitivity: The Fight of Neo-Pentecostal Churches against LGBTQ Rights Organizations over Uganda’s Future"

_religions, doi:10.3390/rel12090707_

Round 1

Reviewer 1 Report

The main arguments in this article are incoherently presented often in a convoluted manner. Some of them come across as contradictory. They make hard work of arguments that could or should be made in a straight forward manner. The author(s) must take a few steps back to isolate and restate their main arguments, ensuring that they are coherent and that they co-relate accordingly. It's a confusing article. I also think the article is too long at more than 20 pages. Tighter writing and straighter argumentation can easily reduce the article to 15 pages. A thorough review of language and grammar is necessary.

Author Response

We appreciate all the suggestions, and we have made the following changes in response to the first reviewer’s recommendations:

  • “The main arguments in this article are incoherently presented often in a convoluted manner. (…) The author(s) must take a few steps back to isolate and restate their main arguments, ensuring that they are coherent and that they co-relate accordingly.”

Our Revisions: Thank you for your suggestions. We have re-evaluated our main arguments and the structure of the study. We decided to extensively reshape especially the introductory part. 

  • "I also think the article is too long at more than 20 pages. Tighter writing and straighter argumentation can easily reduce the article to 15 pages."

Our Revisions: Thank you for your suggestion. Since your opinion was in conflict with that of the second reviewer, we have chosen to follow her/his recommendations and did not shorten our study significantly.

  • "A thorough review of language and grammar is necessary."

Our Revisions: We have addressed all the problems, so we hope we have strengthened our article sufficiently. 

Reviewer 2 Report

The article is clear and convincing. 

The main issue is about whether the thesis is a fact that can be proven, or whether it is offering a way of framing that needs to be taken into account. I would say it is clearly the latter, but it is often presented as if it is the first. This same discrepancy applies to the Culture Wars perspective that the article relies heavily upon - I don't think it is 'proofed' by the one article of Castle as is said on page 6, neither do I think it is the kind of thing that can be proven.

We hear too little about the outcomes of the actual interviews and fieldwork.

The hypothesis is mentioned on page 2 before it is explained. Nor is it explained how and at what stage this hypothesis was formulated. 

The section on religion is the least convincing by far. On page 15 the ritual of traditional religion and Neo-Pentecostals is dubbed identical - this is neither a strange nor new, but it is far from the way it is viewed by the practitioners themselves. The success bias (bandwagon effect?) would apply to progressivists as well if they were winning. 

I miss the discussion of gay perspectives in Africa in general, as for example present in the work of Adriaan van Klinken. That might prove a better local link than the sweeping statements about religion that we find in the article right now. 

Author Response

I appreciate all the suggestions, and I have made the following changes in response to the second reviewer’s recommendations:

  • “The main issue is about whether the thesis is a fact that can be proven, or whether it is offering a way of framing that needs to be taken into account.”

Our Revisions: Thank you for pointing this out. We can see the problem now. We have reconsidered our argument and followed your suggestion that “Culture Wars offer a way of framing that needs to be taken into account.” We have fixed the problem by suggesting that the Castle’s work is a part of an increasing body of evidence which renders the explanatory power of the Culture Wars more plausible (p. 10).  

“We hear too little about the outcomes of the actual interviews and fieldwork.“

Our Revisions: Thank you for pointing this out. We have taken the opportunity to clarify our analysis to emphasize parts where we use the outcomes of the interview (The introductory part and parts numbers 4 and 5).

  • “The hypothesis is mentioned on page 2 before it is explained. Nor is it explained how and at what stage this hypothesis was formulated.”

 Our Revisions: Thank you for correcting our error. We have extensively reshaped the introductory part and explicitly formulated our hypothesis as well as the concept used for our analyses, as you recommended. 

  • “I miss the discussion of gay perspectives in Africa in general, as for example present in the work of Adriaan van Klinken.”

 Our Revisions: Thank you for your recommendation to address Adriaan van Klinken’s work. Unfortunately there just wasn’t space for the general discussion about LGBTQ issues in Africa, so we merely took the opportunity to add an endnote mentioning his last contribution to the field (van Klinken, 2020).

  • “The section on religion is the least convincing by far. On page 15 the ritual of traditional religion and Neo-Pentecostals is dubbed identical - this is neither strange nor new, but it is far from the way it is viewed by the practitioners themselves.”

 Our Revisions: Thank you for this objection. Nevertheless, we respectfully follow the theory of Cognitive Science of Religion (Barrett and Lanman 2008; Slingerland and Bulbulia 2011; Szocik 2017; McCauley et al. 2013; Lawson and McCauley 1990; Boyer 2001; Atran 2002) and the evolutionary framework to explain beliefs and practices concerning supernatural entities. In other words we believe that human cognition can be attracted by forms, despite the manifested content (Slone 2007). Nevertheless, we have taken the opportunity to clarify our concept of religious sensitivity, so it is less easily misread. We hope it fixes the problem.

  • “The success bias (bandwagon effect?) would apply to progressivists as well if they were winning.” 

   Our Revisions: Thank you for this objection. Nevertheless, we argue that it is not the case in Uganda. The conservatives are winning, because of their high moral authority and influence based on local religious sensitivity. Neo-Pentecostal churches successfully deploy features that attract those mental abilities responsible for seeking supernatural reassurance of future avoidance of mishaps (Boyer 2019). We took the opportunity to emphasize this argument in our conclusion.

  • In addition: Once we started revising in response to the peer reviewers’ helpful comments, we also noticed some other problems and revisited several sections so that they were tighter and more to the point.